# Applicability of Maximal Ergometer Testing and Sprint Performance in Adolescent Endurance and Sprint Trained Swimmers

**DOI:** 10.3390/sports9050055

**Published:** 2021-04-28

**Authors:** Adam J. Pinos, Elton M. Fernandes, Eric Viana, Heather M. Logan-Sprenger, David J. Bentley

**Affiliations:** 1Faculty of Health Sciences, Ontario Tech University, 2000 Simoce Street N, Oshawa, ON L1G 0C5, Canada; adam.pinos@ontariotechu.net (A.J.P.); eric.viana@ontariotechu.net (E.V.); Heather.Sprenger@ontariotechu.ca (H.M.L.-S.); 2Canadian Sport Institute of Ontario, 875 Morningside Ave. Suite 100, Toronto, ON M1C 0C7, Canada; efernandes@csiontario.ca

**Keywords:** velocity, metabolism, training, power, time trial

## Abstract

Sprint swimming is a short duration, high intensity sport requiring a relatively greater contribution of energy from anaerobic metabolism. Understanding energy system utilization for the classification of a competitive swimmer (sprint or distance) may be useful for both training prescription and event specialization. The relationship between anaerobic swim ergometer testing and adolescent sprint swimming performance has not been investigated. The purpose of this study was to compare the performance and physiological responses during a maximal all-out ergometer test as well as the maximal anaerobic lactate test in a group of sprint vs. middle-distance specialized swimmers. Sixteen (*n* = 16) competitive swimmers (mean ± standard deviation (SD), age 16.8 *±* 0.7 year; body mass 67.3 *±* 9.8 kg) were categorized into two gender matched groups: sprint (*n* = 8) and middle-distance (*n* = 8). Each athlete performed (1) a 45 s swim ergometer maximal test to determine peak and mean power output (Watts (W)), (2) a MANLT test to determine peak and average velocity as well as the post-exercise lactate response, and (3) a 50 m swim time trial. The sprint group showed a higher mean (*p* = 0.026) and peak (*p* = 0.031) velocity during the MANLT. In addition, blood lactate concentration was significantly (*p* < 0.01) higher in the sprint vs. middle-distance trained group at 3 and 12 min after completion of the MANLT (3-min post 11.29 *±* 2.32 vs. 9.55 *±* 3.48 mmol/L; 12-min post 8.23 *±* 2.28 vs. 7.05 *±* 2.47 mmol/L). The power output during the 45 s all-out swimming ergometer test was higher in the sprint trained group. The results of this study demonstrate the anaerobic contribution to sprint swimming measured during an all-out dryland ergometer test.

## 1. Introduction

Competitive swimming is a unique sport comprising a variety of styles (i.e., freestyle, butterfly, breaststroke, and backstroke), encompassing different combinations and durations, each with contrasting physiological demands and biomechanical considerations [1]. Physiologically, short high-intensity athletic events like of less than 60 s have been shown to result in a significant anaerobic glycolytic contribution to energy metabolism (~2–80%) [1,2,3]. It has been demonstrated that sprint swimming (<100 metres (m) and <50 s) relies on a greater proportion of cellular energy derived via anaerobic glycolysis, as reflected by higher elevations in blood lactate concentration compared to middle (200–400 m and 120–240 s) and longer distance (>400 m) events [1,4,5,6]. In sports performance assessment generally, a variety of testing procedures have been used to establish either aerobic or anaerobic capacity, including oxygen uptake (VO2) during and after exercise (to establish oxygen deficit and debt) and the blood lactate response to exercise or in recovery from all-out exercise [7,8,9,10]. Metabolic analysis during exercise has also been performed in the past using VO2 and blood lactate analysis during incremental tests for determining both aerobic and anaerobic potential [9,11,12,13,14,15]. Consequently, the physiological assessment of all swimmers of different specialization (sprint, middle-, or long-distance) usually occurs with incremental swimming tests with aerobic indices of performance such as the lactate threshold (LT) [9,10].

The maximal anaerobic lactic test (MANLT) was developed to evaluate the changes with training in middle-distance swimmers (specific to the 200 m event distance), while also innately assessing anaerobic capacity [16]. The MANLT consists of four 50 m swims with a 10 s rest between each effort. Blood lactate concentration is measured during 3- and 12-min (min) post-exercise passive recovery in a sitting position, as an indication of the athlete’s maximal anaerobic contribution and rate of lactate clearance [16,17]. Pelayo et al., concluded that the recovery response of blood lactate was significantly related to the in-pool sprint performance of trained para-swimmers [18]. There is currently no known research evaluating the blood lactate response following the MANLT in trained adolescent swimmers and how the performance of the MANLT may differ between sprint and distance-orientated swimmers.

The existing literature supports the benefits of dryland strength and power training for swim sprint performance [19,20,21,22,23,24]. Maximal ‘all-out’ testing has been used in the past to measure peak and average power output in swimmers to determine training progress and exercise prescription [20,23,24,25,26]. Quantifying comparable ‘dryland’ testing in swimming is relevant for further understanding sport-specific training requirements. Understanding energy system utilization for the classification of a competitive swimmer (sprint or distance) may be useful for both training prescription and event specialization [1]. One previous study, assessing dryland power using a swim ergometer with 40 competitive adolescent (15–16-year-old) swimmers, discovered a strong relationship between power and sprint performance (r = 0.91) [26]. However, no studies have directly examined the relationship between the metabolic and performance responses of dryland (swim ergometer) low-duration maximal power testing and in-pool sprint performance in trained adolescent swimmers.

Anthropometric and biomechanical advantages, such as longer limb length and distance per stroke, have been correlated with improved swimming performance in 11–13-year-old swimmers [27]. Swimmers are now beginning to specialize at a younger age; however, there are limited studies comparing the physiological characteristics between sprint and distance specialist swimmers, especially in a ‘young’ developmental age group [1,15,28,29,30]. The studies that have been conducted typically collect physiological data as a secondary measure to either anthropometric or biomechanical measures, which have been shown to contribute more to an athlete’s success at this age [15]. The anaerobic contribution to 100-m sprint swimming in young athletes has been suggested to account for only ~46% of the athlete’s success [1]. As a result of this, there is very little published literature on the definitive anaerobic characteristics of these athletes [1]. Therefore, the purpose of this study was twofold: (1) to evaluate the difference in performance and lactate production in the MANLT and a 45 s all-out anaerobic swim ergometer test between sprint and middle-distance youth swimmers, and (2) to assess the relationship between (sprint) anaerobic performance in the MANLT and the results of swim ergometer testing. Our primary hypothesis was that the power output measured during a swim bench ergometer test would distinguish sprint specialist from middle-distance specialist swimmers. In addition, performance (power output) during the ergometer test would be highly correlated to performance (velocity) of the athletes’ 50 m freestyle sprint.

## 2. Materials and Methods

### 2.1. Subjects

Sixteen (*n* = 16; male = 8, female = 8) well-trained swimmers (mean ± SD, age 16.8 ± 0.7 year; body mass 67.3 ± 9.8 kg) from the same training centre voluntarily participated in this study. All participants were selected for provincial and national events, and this was used as eligibility criteria for the athlete to participate in the study. Participants were recruited through their coach and received written details of the testing procedure prior to participation. Swimmers were categorized into two gender matched groups, sprint (*n* = 8) and middle-distance (*n* = 8), based on competitive swim performances and coaching staff recommendations. The 50-m freestyle sprint performance was utilized as the differentiator for groups. The study was approved by the Research Ethics Boards at both Ontario Tech University and Canadian Sport Institute Ontario. The subjects were informed of the benefits and risks of the investigation prior to signing an institutionally approved informed consent document to participate in the study. Informed consent was obtained from all subjects involved in the testing.

### 2.2. Procedures

Each participant underwent a preliminary anthropometric assessment, measuring height in centimeters (cm) and body mass in kilograms (kg). All participants completed three performance tests, depicted in Figure 1, two of which were conducted in the pool and one was a laboratory-based assessment. Athletes completed a similar overall training load (volume and intensity) twelve weeks prior to testing; however, the sprint and middle-distance groups received separate training programs structured for their events. The in-pool testing consisted of two tests: (1) 50 meter (m) time trial to determine average sprint velocity and (2) 4 × 50 m maximal anaerobic lactate test (MANLT) [16,18]. The MANLT consisted of four consecutive 50 m sprints with 10 s rest between each effort [16,18]. The laboratory testing comprised an ‘all-out’ 45-s sprint test on a stationary swim ergometer (VASA, Essex Junction, VT, USA), measuring average power over the course of the 45 s [25,26]. All tests occurred on separate days over a two-week period with a minimum of 48 h between tests. Athletes completed the same standardized warm up protocol before every test session consisting of a 10 min dry-land warm-up followed by a 10 min in-pool warm-up consisting of both general swimming and skill-focused drills such as kicking and pulling. All pool testing was conducted in a 50 m pool. The MANLT began with a push start from the wall with one hand and two feet on the wall to start [16,18]. The 50 m time-trial started from a block using a dive start and commenced with an auditory cue. Each participant completed the swimming testing using the same stroke, front crawl.

### 2.3. Maximal Anaerobic Lactate Test (MANLT)

Each 50 m sprint time and stroke rate were recorded for the MANLT. At the end of the last 50 m effort, the athlete’s heart rate (HR) was immediately measured using a pulse oximeter (Nonin Medical, Plymouth, MN, USA). Upon completion of the fourth 50 m, the athlete was assisted from the pool and rested in a seated position. A 3- and 12-min post-exercise blood lactate (BLa) sample was then obtained from them by puncturing the fingertip under aseptic conditions with an automatic lancet, and approximately 25 µL of blood was analyzed using a portable hand-held blood lactate analyzer (EDGE, Warszawa, Poland) [16,18].

### 2.4. Swim Ergometer Test

Participants performed a 45 s all-out swimming test using a stationary swim ergometer (VASA, Essex Junction, VT, USA) [26]. Prior to testing, participants completed a standardized ten-minute dynamic dryland warm-up, similar to the dryland warmup completed on the swim testing day. Upon completion of the warmup, participants performed a 45 s maximal sprint on the isokinetic swim bench. No resistance was used to ensure that every athlete could pull with maximal velocity for the entire test. The testing protocol was used based on a previous study showing the relationship between anaerobic swim bench power and in-pool sprint performance [25,26]. Power output was recorded every stroke and was subsequently averaged every 5 s and for the entire trial. The peak power (W) was determined as the highest 5 s average period. The 3- and 12-min post exercise BLa samples were obtained using the method previously described.

### 2.5. 50 m Time Trial

On a separate testing day from the MANLT test, participants completed a 50 m freestyle time trial. The testing environment was under simulated racing conditions and participants wore racing suits and were paired with a swimmer similar in speed to swim against. Before the trial, participants completed the same warm up as the MANLT. Participants started each trial with a dive start initiated by an auditory cue from the swimming starting device (Startime III, Swiss Timing, Corgemont, Switzerland). Upon completion of the 50 m swim, heart rate was immediately obtained using a pulse oximeter (Nonin Medical, Plymouth, MN, USA). At 3- and 12-min post, swimmers’ blood lactate was measured as previously described. The average swim velocity (m/s) for the 50 m effort was determined using Kinovea video analysis software (Kinovea Software, Boston, MA, USA). The time started when the visual cue of a flash from the starting device appeared in the video, and the end was when the participant touched the wall.

### 2.6. Statistical Analyses

Data analysis was performed using SPSS 26 (IBM Analytics, Armonk, NY, USA) statistical software. Mean and standard deviation (SD) (95% confidence interval) were calculated for each measure in each group (sprint vs. distance). The sample was assessed for normalcy using a Shapiro–Wilk test and a value *p* > 0.05 was reported, determining the data were normally distributed. One-way ANOVAs were tested to establish the difference (*p* < 0.05) between sprint vs. middle-distance groups for each measure. A Pearson correlation was utilized, with a priori *p*-value of less than 0.05 to examine the relationship (r) between testing 50 m performance and each performance measure, determining confidence intervals (CI) and effect sizes (ES) for each.

## 3. Results

### 3.1. Maximal Anaerobic Lactate Test (MANLT) 

The performance results from the MANLT and time trial performance are reported in Table 1. The mean and peak velocity (m/s) during the MANLT were significantly (*p* < 0.05) higher in the sprint versus middle-distance group, respectively (mean velocity: 1.57 ± 0.06 vs. 1.49 ±0.07; peak velocity 1.66 ± 0.07 vs. 1.57 ± 0.07; ES = 1.29 and 1.23). The blood lactate (BLa) concentration at 3 and 12-min post MANLT was also not significantly higher in the sprint versus middle-distance group, respectively (3-min post 11.01 ± 2.32 vs. 9.55 ± 3.48 mmol/L; 12-min post 8.07 ± 2.28 vs. 7.05 ± 2.47 mmol/L, ES = 0.49 and 0.43). The 50 m time trial mean velocity was significantly higher (*p* > 0.05) in the sprint versus middle-distance group, respectively (1.86 ± 0.10 vs. 1.72 ± 0.08, ES = 1.44). The BLa concentration at 3 and 12-min post 50 m time trial was significantly higher in the sprint versus middle-distance group, respectively (3-min post 9.88 ± 2.12 vs. 6.31 ± 2.77 mmol/L, *p* < 0.05; 12-min post 8.09 ± 1.07 vs. 4.70 ± 1.58 mmol/L, *p* < 0.01; ES = 1.45 and 2.51).

### 3.2. Swim Ergometer Test

The absolute mean power output at 10, 30 and 45 s of the 45 s all-out swim ergometer test is shown in Figure 2. The absolute mean power during the 45 s test was not significantly higher in the sprint versus middle-distance group, respectively (174.9 ± 36.9 vs. 150 ± 30.9 (W), ES = 0.73). The sprint group generated a higher absolute power output throughout the trial; however, the absolute mean power in the first 10 s and 30 s during the swim ergometer test was also not significantly higher (ES = 0.65 and 0.69) in the sprint versus middle-distance group, respectively. Relative mean and peak power outputs (W/kg) at 10, 30, and 45 s between groups were also not significantly different (ES = 0.56, 0.59, 0.64, 0.75) between groups. The lactate concentration at 3 and 12-min post swim ergometer test was significantly higher in the sprint versus middle-distance group, respectively (3-min post 8.1 ± 2.17 vs. 5.85 ± 1.32 mmol/L, *p* < 0.05; 12-min post 5.3 ± 1.29 vs. 3.8 ± 0.79 mmol/L, *p* < 0.05, ES = 0.49 and 0.43).

### 3.3. Correlations with 50 m Sprint Performance

Significant correlations were found between the peak and mean velocity of the MANLT and 50 m time trial performance (r = 0.82 and 0.81, *p* < 0.01), and can be found in Table 2. No relationship was found between the BLa of the MANLT and 50 m performance. However, 3 and 12-min post BLa from the 50 m time trial, respectively, was correlated with 50 m performance (r = 0.54, *p* < 0.05; r = 0.68, *p* < 0.01). Absolute 10, 30, and 45 s mean and peak power, respectively, were significantly correlated with 50 m performance (r = 0.79, 0.83, 0.86, 0.90; *p* < 0.01), and can be seen in Table 2 and Figure 3. However, only relative mean and peak 45 s power outputs, respectively, were weakly correlated with 50 m velocity (r = 0.51, *p* < 0.05; r = 0.68, *p* < 0.01).

Mean absolute 10, 30, 45, and peak power at 45 s, respectively, were strongly correlated with peak MANLT velocity (r = 0.75, 0.77, 0.79, 0.75; *p* < 0.01). Relative power measures of the 45 s test, however, were weakly correlated with peak MANLT velocity (r = 0.54, 0.56, 0.58, 0.60; *p* < 0.05). Similar to peak MANLT velocity, mean MANLT velocity was more strongly correlated to all absolute mean power outputs of the 45 s test; however, all relative power outputs were also strongly correlated with mean MANLT velocity (r = 0.74–0.82; *p* < 0.01).

A strong correlation was observed between 3 min BLa and peak MANLT velocity (r = 0.64, *p* < 0.01). Correlations were also observed for absolute 10, 30, 45 and relative 10 and 30 s mean power outputs and 3-min BLa (r = 0.61, 0.57, 0.53, 0.54, 0.50; *p* < 0.05). Additionally, when examining the relationship between 3-min BLa response between tests, 45 s ergometer results did not correlate (r = 0.39) with MANLT, but the MANLT and ergometer 3-min BLa, respectively, correlated with time trial BLa response (r = 0.85 and 0.64, *p* < 0.01).

## 4. Discussion

The purpose of this study was to determine the difference in performance in the MANLT and 45 s all-out anaerobic swim ergometer test between sprint and middle-distance swimmers. Furthermore, this study examined the relationship between (50 m sprint) anaerobic performance and the MANLT as well as the results of swim ergometer testing. The results of this study demonstrate that sprint trained swimmers performed better in each performance test compared to a group of middle-distance focused swimmers. Likewise, the lactate recovery physiological response to the testing was also different between the two specialist groups, indicating a unique physiological adaptation in the groups to their respective training. To support this, strong relationships (r = 0.79–0.90) were observed between the 45 s max ergometer test, MANLT, and 50 m performance.

The physiological demands of sprint and middle-distance swimming have been previously quantified [1,4,5]. Whilst swimming generally requires athletes to minimize drag, sprint swimming is a unique discipline in that it requires athletes to exert a high level of force and recruit a greater proportion of fast twitch muscle fibers [1], but physiologically requires lower total absolute anaerobic contribution than middle-distance swimming due to the shorter duration and subsequent decreased overall metabolic demand [31]. The strong relationship between the 45 s ergometer test and the 50 m pool time trial performance indicates a positive relationship between a more general laboratory method for assessing anaerobic upper body power in swimmers and in-pool sprint performance. That said, other studies have shown lower limb power is also important at the start of a sprint swimming event, enabling athletes to perform an optimal starting portion of the race. Therefore, whilst there were strong correlations between power output during the ergometer test and 50 m pool performance, other tests of lower limb power should also be included in a battery of performance tests.

Whilst controversial, previous studies have shown that dryland strength and power training may improve swimming performance [32,33]. There is very little research published on the relationship between swim ergometer testing and sprint performance. One older study found there was a weak relationship with 25-yard sprint performance, but the researchers used an older population [26]. In this study, we found significant correlations between relative and absolute power output achieved in an anaerobic ‘all-out’ swim ergometer test and 50 m sprint performance. No significant differences were observed between sprint and middle-distance groups for the ergometer test. However, this study determined that there were significant differences in mean and peak velocity between the sprint and middle-distance group, and strong correlations between absolute power outputs and 50 m velocity along with mean and peak MANLT velocity. This may indicate a relationship between upper body power achieved on a dryland test and its significant applicability to in-pool sprint performance. It is interesting to note that correlations between relative swim ergometer power outputs with time trial and MANLT are much stronger with mean MANLT velocity (r = 0.74–0.79, *p* < 0.01) than peak MANLT (r = 0.54–0.60, *p* < 0.05) and time trial velocity (r = 0.43–0.68, *p* = 0.1–0.01). The evidence from this study suggests that this test could be useful for tracking the development of upper body absolute power in sprint swimmers and could be used to track performance or for talent identification purposes. It is clear, however, that strength or power alone does not determine swimming velocity, and there are many factors that need to be taken into account when trying to assess contributing factors to a swimmer’s performance, such as start reaction time, lower limb power, swimming biomechanics, and anthropometry [23,27]. A number of studies have explored the relevance of dryland anaerobic power using a swim ergometer on swim performance generally [23,24,25,26,27]. However, the significance of anaerobic power, especially all-out anaerobic power using a swim ergometer on sprint swimming performance, is limited [25,26]. Future work should aim to examine the effect an ergometer versus in-pool training has on swimming performance.

Anaerobic glycolysis has also been found to contribute ~55% of metabolic energy (adenosine triphosphate) production during 100-m and 200-m (sprint) running events, these distances being similar in duration to the 50-m and 100-m sprints in swimming [34,35]. In support, the results of this study found a greater blood lactate (mmol/L) accumulation in the sprinting population post-test in comparison to the middle-distance group (11.29 ± 2.32 vs. 9.55 ± 3.48). This could be a result of higher recruitment of type 2 muscle fibers from the sprinters, causing greater glycolytic metabolism and lactate accumulation. A close relationship (r = 0.89) has also been previously found between post-exercise blood lactate concentration and average velocity maintained over 400 m and 800 m in trained middle-distance runners during competition, with durations of 3 min or less, similar to that of the middle-distance events in swimming [34,35]. In this study, a significant correlation was found with 50 m sprint performance and the average velocity during the MANLT (~2 min). Sprint swimming is shorter in duration and is highly explosive in nature, therefore requiring less energy production overall but at a faster rate in comparison to middle-distance swimming. As such, the sprint group showed a greater anaerobic lactate response in comparison to the middle-distance group, which indicates a greater flux through glycolysis and more conversion of pyruvate to lactate. Moreover, the average velocity and post-exercise lactate response in the MANLT were originally designed to monitor performance in para-athletes [16]. Pelayo and colleagues later demonstrated that the blood lactate recovery to the MANLT could also be tested, and the recovery improvements provided practical and useful criteria to monitor the effects of training in able-bodied middle-distance swimmers [18]. In this study, both the lactate production and performance of the MANLT were unique to each swim discipline. Therefore, the results of the present study support the idea that performance and blood lactate responses can be used to distinguish aerobic and anaerobically specialized swimmers. As such, the results support the notion that the MANLT may be used as a valid anaerobic test of swimming performance in a competitive adolescent population. Additionally, if facilities are not available for MANLT testing, quantifying performance using a swim ergometer test may be a simple yet valid assessment tool to compliment other assessment strategies and overall performance of adolescent swimmers, not only to monitor gym-based training adaptations, but also for talent identification purposes and as a surrogate measure of in-pool performance. As well, the results from this study indicate the applicability of the MANLT for a younger able-bodied population. This research had some limitations, particularly the specialization of the sample participants and the sample size. At their age, many competitive swimmers compete in a variety of events, and selection criteria were left to the coach’s discretion. Additionally, the homogenous training group this sample of participants was recruited from was small (~20 swimmers), making it difficult to have a large enough sample size for strong statistical conclusions to be made.

## 5. Conclusions

The results of this study highlight the importance of the MANLT and 45 s max swim ergometer tests as a tool to monitor the anaerobic performance capacity of adolescent swimmers. Moreover, the results of the 45 s swim ergometer maximal test and MANLT may be used to distinguish between sprinters and middle-distance swimmers. In addition, the 45 s swim ergometer test could be used as an alternative to assess swimmers’ anaerobic capacity outside the pool. Future research should examine the effects of swim ergometer sprint training on sprint swim performance.

## Figures and Tables

**Figure 1 sports-09-00055-f001:**
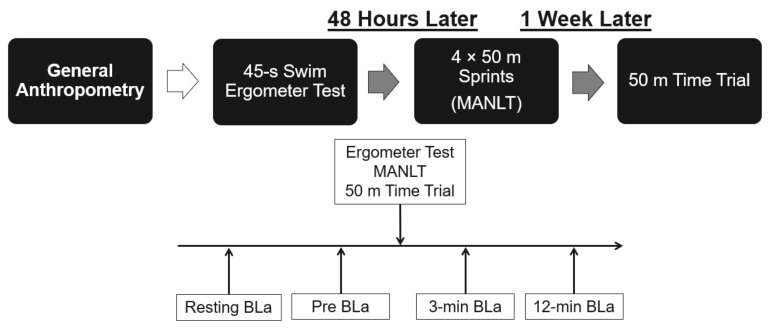
In-pool and dryland testing schedule with blood lactate concentration (BLa) timing for testing presented above.

**Figure 2 sports-09-00055-f002:**
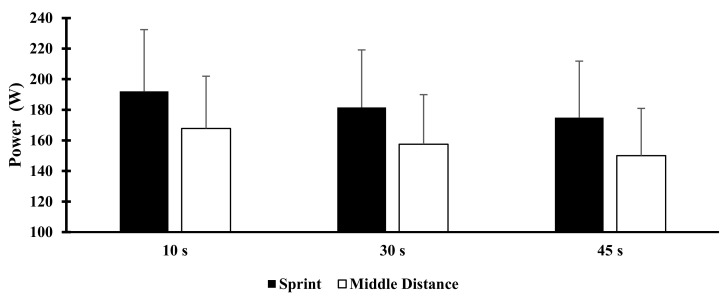
Forty-five-second (s) maximal ergometer test mean power at 10, 30, and 45 s in both the sprint (S, *n* = 8) and middle-distance (MD, *n* = 8) groups. The sprint (10 s: S: 192.2 ± 40.3 vs. MD: 167.8 ± 34.1 Watts (W); 30 s: S: 181.6 ± 37.6 vs. MD: 157.5 ± 32.3 W; 45 s: S: 174.9 ± 36.9 vs. MD: 150 ± 30.9 W). Values are means ± SD.

**Figure 3 sports-09-00055-f003:**
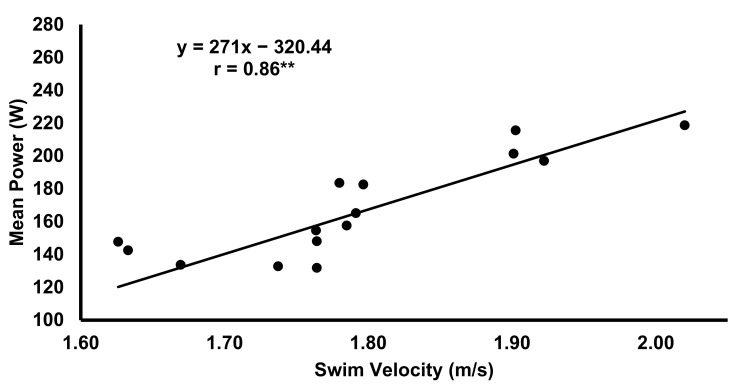
Correlation between mean power of the 45-s maximal ergometer test and 50-m freestyle velocity (r = 0.86, *p* < 0.01). ** significance (*p* < 0.01).

**Table 1 sports-09-00055-t001:** Differences in peak and mean velocity (m/s) of the maximal anaerobic lactate test (MANLT), sprint velocity and post-exercise blood lactate concentrations after the 50 m (m) time trial for sprint (*n* = 8) and middle (*n* = 8) distance swimmers. Cohen’s D effect sizes (ES) between groups and Pearson r correlation represented below.

Variable	Sprint	Middle Distance	ES	Correlation with 50 m
Peak velocity (m/s)	1.66 ± 0.07 (1.60–1.72)	1.57 ± 0.07 (1.51–1.63)	1.29	0.82 **
Mean velocity (m/s)	1.57 ± 0.06 (1.52–1.62)	1.49 ± 0.07 (1.43–1.55)	1.23	0.81 **
Time trial blood lactate: 3 min post (mmol/L)	9.88 ± 2.12 (8.10–11.65)	6.31 ± 2.77 (4.00–8.62)	1.45	0.54 *
Time trial blood lactate: 12 min post (mmol/L)	8.09 ± 1.07 (7.19–8.99)	4.70 ± 1.58 (3.38–6.02)	2.51	0.68 **
Time trial mean velocity (m/s)	1.85 ± 0.10 (1.76–1.93)	1.72 ± 0.08 (1.65–1.78)	1.44	-

Data are mean ± standard deviation (SD), followed by confidence interval (CI) in brackets. * Correlation is significant at *p* < 0.05. ** Correlation is significant at *p* < 0.01.

**Table 2 sports-09-00055-t002:** The peak and mean power (Watts (W)) and 3 and 12-min post blood lactate (BLa) concentrations of the 45-s swim ergometer test for both sprint (*n* = 8) and middle-distance (*n* = 8) swimmers.

Ergometer Result	Sprint	Middle Distance	Correlation between Ergometer Result and 50 m
Mean 10 s power (W)	192.2 ± 40.3	167.8 ± 34.1	0.79 **
Mean 30 s power (W)	181.6 ± 37.6	157.5 ± 32.3	0.83 **
Mean 45 s power (W)	174.9 ± 36.9	150 ± 30.9	0.86 **
Peak 45 s power (W)	216.4 ± 51.3	182.4 ± 43.2	0.90 **
3-min Post BLa (mmol/L/L)	8.1 ± 2.17	5.85 ± 1.32	0.50 *
12-min Post BLa (mmol/L/L)	5.3 ± 1.29	3.8 ± 0.79	0.47

Data are mean ± standard deviation (SD). * Correlation is significant at *p* < 0.05. ** Correlation is significant at *p* < 0.01.

## Data Availability

Data available on request due to sport related restrictions on data sharing. The data presented in this study are available on request from the corresponding author. The data are not publicly available due to the sensitivity of the data related to sport performance.

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
