# Peer review of "Applicability of Maximal Ergometer Testing and Sprint Performance in Adolescent Endurance and Sprint Trained Swimmers"

_sports, 2021, doi:10.3390/sports9050055_

Round 1

Reviewer 1 Report

Abstract

Dear authors 

First of all, I want to thank the authors for submitting their work to the journal Sports. I think the authors did a good job overall. However, there are some topics that need to be revised and improved in order to increase the quality of the work. This article addresses a problem that is not often discussed in the international scientific literature. The data presented here are of particular interest to sports science researchers and, specifically, to coaches and swimmers, in order to improve their performance.

 Abstract

Page 1 – Line8 - please try to insert a short introductory paragraph at the beginning of the abstract

Introduction

Page 1 – Line 24 - add the acronym i.e., (i.e., freestyle, but-24 terfly, breaststroke, and backstroke)

Page 1 – Line 27 - You can improve the connection between these two paragraphs, in order to make the transition clearer.

Page 1 – Line 24 to 40 - All the content placed in this initial paragraph is pertinent, however the author should try to promote a better interconnection between the ideas so that the topics that he is inserting have some direct connection with the idea immediately iterior. These changes would be very useful for the comprehension and fluidity of the text.

Pag 2 – Line 68 - although there are few investigations, it is essential to describe what they found, and to cross that information between studies. In this way the author will provide the reader with a current view of what has already been written on the topic, creating the window of opportunity for the need to develop further studies.

Pag 2 – Line 70 - Please provide the study hypotheses.

In general, the introduction can be improved. Improve the link between ideas, mainly in the first phase of the introduction and fundamental. Although the information collected is framed with the topic, I believe that the last paragraph regarding the definitive anaerobic characteristics oh young swimmers can be greatly improved and that this can help to increase the understanding of the study problem.

Material and Methods

Subjects

Pag 2 – Line 77- please identify the gender of the sample.

Pag 2 – Line 81 -Please provide more details on how to categorize into groups. In addition, it would be good if there was a reference of support for the selection protocol.

Pag 2 – Line 92 – please add reference to anthropometric assessment

Pag 2 – Line 90 – 107 - Please add references that support the described procedures, this point is essential to increase the reliability of the study

Maximal anaerobic lactate test (MANLT)

Pag 3 – Line 109 – 116 – The same, please add references that support the described procedures, this point is essential to increase the reliability of the study

Pag 3 – line 129 - 50m time trial – one more time, please add references that support the described procedures, this point is essential to increase the reliability of the study

Pag 4 - Table 1 - check for missing values

Pag 6  - Table 2 - please add a caption for all the abbreviated terms in the table

Results

This section is properly structured. Good work.

Discussion

Pag 7 -Line 243 - Please indicate which studies you refer to in this paragraph

Conclusions

Pag 8 – Line 306 - I believe that you can develop a little more the conclusions resulting from your study and its applicability in the practical context, as well as future lines of research.

References

The references used are adapted for the research topic. However, there is a high number of references that are too old. You should try to replace them if possible.

Reviewer 2 Report

Thanks for submitting the manuscript. The findings of this study may potentially contribute to the swim training and talent identification purposes. However, there are several issues need to be addressed by the authors:

1) The title did not reflect the major findings of this study. The authors may consider to revise it.

2) Abstract: The results of the study is not clear and include some typos, such as “’”, please revise it.

3) Materials and Methods: Please provide the details of whether the research team obtained the informed consent from the participants’ parent/guardian (especially for those who are under 18)?

4) It will be easier for the readers to follow the research design and procedure (p. 2-3) with a diagram.

5) Please clarify is there any data from the baseline study? It will be more persuasive to show the longitudinal data.

6) The hypothesis of this study is not clear. The authors need to spell out the underlining research questions and hypothesis in the introduction section.

7) Please provide more justifications for the selection of statistical analysis methods. Using ANOVA and correlation on study with small sample may provide a less meaningful result. Some literature is suggesting to use Bayesian approach (Chekouo et al., 2020; Sakshaug, Wisniowski, Ruiz, & Blom, 2019), what is your response?

8) The authors also need to report their limitations in the discussion section.

References

Chekouo, T., Stingo, F. C., Class, C. A., Yan, Y. Q., Bohannan, Z., Wei, Y., . . . Do, K. A. (2020). Investigating protein patterns in human leukemia cell line experiments: A Bayesian approach for extremely small sample sizes. Statistical Methods in Medical Research, 29(4), 1181-1196. doi:10.1177/0962280219852721

Sakshaug, J. W., Wisniowski, A., Ruiz, D. A. P., & Blom, A. G. (2019). Supplementing Small Probability Samples with Nonprobability Samples: A Bayesian Approach. Journal of Official Statistics, 35(3), 653-681. doi:10.2478/jos-2019-0027

Reviewer 3 Report

Dear Authors,   You made your point well in this manuscript, but you do not fully reflect your point in the title of the paper. Could you say "Proof of Applicability of . . .", or some other words borrowed from your Discussion, manuscript lines 299 - 304 ? 

To me it seems that you have in your paper https://pubmed.ncbi.nlm.nih.gov/33561823/ an early pointer to current result (or a stepping up paper to current one?) of assessing a sprint ability by a training modality on land. Isn't it better to also refer to this previous paper?   Lines 59 - 61 forward a clear statement about your main point. Could this be used earlier, in the Abstract, for instance?   The figure 1 could be improved - in my opinion - by putting the values in a 2D display (the idea is borrowed from Physics):   The sprint scores at the Horizontal axis
The Middel Distance at the Vertical axis   Then: every value of the Sprint is below the neutral axis y=x, because the Sprints have scores >  Middle Distance scores.

[I drew a plot for you, but this MDPI editor software does not allow to put it in here.] The final check for  removing typos has still to be done. A few are listed below:  
  1. inconsistent shortcut of Watt. you mixed lower case & upper case. The 1st occurrences are at lines 13, 127 . . .;
  2. at line 43 you refer to formula (16), should this be reference [16]?
  3. at line 47, 48 you redundantly refer twice to [18] in one sentence; such double reference  in one sentence also happens at lines 66, 67; also erroneous double mentioning is 'second (sec)' and 'minute (min)' at lines 97, resp. 101;
  4. lines 53, 55 use single and double apostrophes. Does this inconsistency happen further?
  5. lines 104, 105 have a wrong sentence with twice both the words 'wall' and 'start';
  6. line 69 misses a word about: '... little published definitive anaerobic ...'
  7. lines 70, 72 have inconsistent list numbers:  1), resp. [2], shuld be 1), resp. 2);
  8. line 77 should have blanks before and after the first plus-minus sign;
  9. line 96 is too cryptical about the MANLT test, while it is explained more fully later. Please change the order;
  10. at line 111 you say HR. I don't understand it. Is it heart rate?
  11. spelling errors: line 149 'vale' stead of 'value';
  12. line 157 mentions for the 1st time 'ES'.  What is it? It occurs often.
  13. Table 2 has poor layout, the 12 minute Post-Bla mmol/L/L is split over two lines;
  14. Figure 2 says in the capture twice r =0.86 and twice p<0.01. Why not both once in the figure and no repetition of r=0.86 (p<0.01) in the capture?
  15. the end of the Conclusion, at lines 310 - 311 repeats the end of the Discussion.  I recommend to delete lines 310 - 311;
  16. lines 314 - 318 contain a number of serious slips and errors.

Round 2

Reviewer 2 Report

Thanks for submitting the revised manuscript to address my concerns. I am fully satisfied with the responses and changes made by the authors.